# Biomedical Polyurethanes for Anti-Cancer Drug Delivery Systems: A Brief, Comprehensive Review

**DOI:** 10.3390/ijms23158181

**Published:** 2022-07-25

**Authors:** Marcin Sobczak, Karolina Kędra

**Affiliations:** 1Department of Biomaterials Chemistry, Chair of Analytical Chemistry and Biomaterials, Faculty of Pharmacy, Medical University of Warsaw, 1 Banacha St., 02-097 Warsaw, Poland; 2Military Institute of Hygiene and Epidemiology, 4 Kozielska St., 01-163 Warsaw, Poland; 3Institute of Physical Chemistry, Polish Academy of Sciences, 44/52 Kasprzaka St., 01-224 Warsaw, Poland; kkedra@ichf.edu.pl

**Keywords:** biomaterials, drug delivery systems, anti-cancer drug delivery systems, biomedical polyurethanes, biodegradable polyurethanes, polyurethane chemistry

## Abstract

With the intensive development of polymeric biomaterials in recent years, research using drug delivery systems (DDSs) has become an essential strategy for cancer therapy. Various DDSs are expected to have more advantages in anti-neoplastic effects, including easy preparation, high pharmacology efficiency, low toxicity, tumor-targeting ability, and high drug-controlled release. Polyurethanes (PUs) are a very important kind of polymers widely used in medicine, pharmacy, and biomaterial engineering. Biodegradable and non-biodegradable PUs are a significant group of these biomaterials. PUs can be synthesized by adequately selecting building blocks (a polyol, a di- or multi-isocyanate, and a chain extender) with suitable physicochemical and biological properties for applications in anti-cancer DDSs technology. Currently, there are few comprehensive reports on a summary of polyurethane DDSs (PU-DDSs) applied for tumor therapy. This study reviewed state-of-the-art PUs designed for anti-cancer PU-DDSs. We studied successful applications and prospects for further development of effective methods for obtaining PUs as biomaterials for oncology.

## 1. Introduction

According to the World Health Organization, the number of cancer cases is expected to reach 22 million per year by 2035 [1]. Surgery, radiotherapy, and chemotherapy are the most common cancer treatments. Chemotherapy is the most frequently used systemic treatment for suppressing cancer cell proliferation, disease progression, and metastasis. However, chemotherapeutic drugs also inevitably attack normal cells, causing dangerous adverse effects. Therefore, new anti-cancer drug delivery systems (DDSs) that maintain or improve the efficacy of chemotherapy while reducing the severity of reactions and side effects are urgently needed [2,3,4,5,6].

DDSs are a particular type of biomaterials. Biomaterials are defined as any natural, semi-synthetic, or synthetic substances engineered to interact with biological systems in order to direct medical treatment or diagnostics. These materials must be biocompatible, meaning they perform their function with an appropriate host response. Biomaterials can be generally divided into the following groups: polymers, metals, ceramics, carbon materials, and various composites [5,7].

Biodegradable or bioresorbable polymers are of utmost interest since these biomaterials can be broken down, excreted, or resorbed without removal or surgical revision. One of the most promising groups of biomedical and biodegradable polymers is aliphatic or cycloaliphatic polyurethanes (PUs) [8,9]. PUs are popular because of their segmented-block structure, which endows them with a broad range of versatility in terms of tunable mechanical, physicochemical, and biological properties, as well as blood and tissue compatibility and their biodegradability. PUs are characterized by valuable and unique physicochemical properties that are very important for biomedical applications. The soft segments determine the low glass-transition temperature (T_g_) or high elasticity of PUs. The hard segments determine the high T_g_, melting point (T_m_), or high strength of PUs. The values of fail stress and elongation at break are 5–230 MPa and 200–1300%, respectively. PUs are also characterized by corrosion resistance, high gloss and weathering resistance [10,11,12,13,14].

PUs have traditionally been used as biostable and inert materials in catheters, heart valves, prostheses, and vascular grafts [10,11,12,13]. However, interest in designing resorbable/degradable PUs for tissue engineering and drug delivery systems (DDSs) has also been increasing in recent years [10,14,15,16,17,18,19,20].

Biomedical PUs (BioPUs) can be synthesized by incorporating hydrolyzable segments into their backbones (e.g., polyether, polyester, and polyamide segments). A second strategy for making such PUs is to use amino-acid-derived diisocyanates and biocompatible aliphatic or cycloaliphatic diisocyanates (ICs) in the synthesis. These components have lower toxicity than traditional ICs, such as 4,4′-methylenebis(phenyl isocyanate) (MDI) and toluene 2,4-diisocyanate (TDI). An additional benefit of such PUs is their proven ability to promote cell adhesion and proliferation without adverse effects [10,13,15,19].

Recent developments of BioPUs as short-, medium-, or long-term anti-cancer DDSs are described in detail in this review.

## 2. Synthesis and Properties of Biomedical Polyurethanes Used in Anti-Cancer Drug Delivery Systems

BioPUs can be obtained through the polyaddition or polycondensation processes. The polyaddition process involves ICs reacting with bi- or multi-functional polyols, polyamines, alcohols, and amines. BioPUs are obtained in a one- or two-step method (prepolymer method). In the first step of the second method, polyols are continuously stirred with ICs, and the obtained prepolymers are then extended using chain extenders (such as a diol or diamine) (Figure 1). Different types of ICs (aromatic, aliphatic, and cycloaliphatic) are used in BioPU synthesis: TDI, MDI, L-lysine diisocyanate ethyl ester (LDI), hexamethylene diisocyanate (HDI), isophorone diisocyanate (IPDI), bis(2-isocyanatoethyl) disulfide (CDI), dicyclohexylmethane 4,4′-diisocyanate (HMDI), and tetramethylene diisocyanate (BDI) (Figure 2). Polyols are used polyesters (e.g., poly(ε-caprolactone) (PCL) and polylactide (PLA)), polyethers (e.g., poly(ethylene glycol) (PEG) and poly(propylene glycol) (PPG)), and polycarbonates (poly(trimethylene carbonate) (PTMC), poly(ester-carbonate)s, and copolymers of cycle monomers (e.g., glycolide (GG) and lactide (LA)) (Figure 3). The polyols are usually prepared using ring-opening polymerization (ROP) of heterocyclic monomers (e.g., esters and carbonates) in the presence of cationic or anionic initiators, enzymes, and coordination catalysts. Chain extenders are often used, such as 1,4-butanediol (BDO), diethylene glycol (DEG), ethylene glycol (EG), 1,3-propanediol (PDO), 1,4-diaminobutane (1,4-DAB), 1,2-diaminoethane (1,2-DAE), 1,6-diaminohexane (1,6-DAH), and 1,8-diaminooctane (1,8-DAO). The most popular polyaddition catalysts are 1,4-diazabicyclo-[2.2.2]-octane (DABCO), dibutyltin dilaurate (DBTDL), dibutyltin dioctanate (DBTDO), and tin(II) 2-ethylhexanoate (SnOct_2_).

It is also worth mentioning that isocyanate crosslinker molecules contain aromatic phenolic groups (e.g., functionalized catechol, processed lignin, and phenolic compounds from tannin). It was found that some PUs obtained from these substrates do not adversely affect cell proliferation [21].

There are also other less frequently used methods of obtaining PUs, which are discussed later in this article.

BioPUs are biodegradable or non-degradable polymers whose biocompatible characteristics can be tailored to biological systems, such as those of the blood, organs, and tissues, and are biodegradable depending on their components [20]. PUs’ chains comprise relatively long and flexible polyols (soft segments) and a relatively rigid part imparted by chain extenders and ICs (hard segments). PUs are unique polymeric materials with a wide range of physical and chemical properties. The mechanical properties of PUs can easily be modified by altering the soft-to-hard segment ratio and composition [20,22].

The physical properties of BioPUs should be adequate for particular medical applications. Implantology requires materials with optimal yield modulus and strength, along with fatigue, wear, or friction resistance. The modulus, mechanical strength, and fatigue resistance are essential for PUs to be used in reconstructive surgery of soft tissue and cardiovascular. In contrast, the mechanical strength, modulus, and thermal expansion along with conductivity, wear, and abrasion resistance all affect the performance of dental materials. The properties of PUs are usually controlled by the structure, degree of crystallinity, molecular weight, soft-to-hard segment ratio, number of crosslinks, pendant groups, additives, surface properties, etc. All these factors will also affect the PUs’ biocompatibility to some extent [22,23,24,25].

PUs have several properties required from synthetic biomaterials. They can be reproduced as pure materials, fabricated into the desired form without being degraded or adversely changed, and sterilized without changing properties or form. Moreover, PUs have no physical, chemical, or mechanical properties that are adversely altered by the biological environment unless they were purposely designed as degradable materials. PUs also have no adverse effect on the recipient of the implant. These biomaterials have neither induced thrombosis or abnormal intima formation nor interfered with normal clotting mechanisms. PUs do not lead to cell fragility or aging, allergic reactions, hypersensitivity, or carcinogenic, mutagenic, teratogenic, or toxic reactions [22,23,24,25].

## 3. Polyurethane Anti-Cancer Drug Delivery Systems

As mentioned earlier, one of the intensively developed directions of pharmacy research is anti-cancer polyurethane DDSs (PU-DDSs). PU-DDSs provide stable formulation, improved pharmacokinetics, and a degree of ‘passive’ or ‘physiological’ targeting to tumor tissue. To date, several kinds of anti-cancer PU-DDSs have been developed [10,13,26]. The developed carriers contain cytostatic drugs such as cyclophosphamide (CYCLOPHO), doxorubicin (DOX), epigallocatechin gallate (ECG), 5-fluorouracil (5-FU), gefitinib (GEF), methotrexate (METX), temozolomide (TMZ), and paclitaxel (PACL) (Figure 4). These drugs are commonly known and characterized with different mechanisms of pharmacology actions [27,28,29,30,31,32,33,34]. Various types of anti-cancer PU-DDSs have been prepared (Figure 5) (Table 1).

The controlled release of a drug is a crucial property of DDS due to the effectiveness and biosafety of the therapy. Considering the physical and chemical properties of both the drug and the matrix, the mechanisms used are diffusion, erosion, swelling, or osmosis. However, most often a mixed mechanism is used (Figure 6) [6,25]. Diffusion is a concentration-gradient-driven mass transfer process. In a diffusion release system, the drug’s diffusion kinetics is the rate limiting step. In a diffusion-mediated controlled release system, the drug can either be dispersed or dissolved in the polymeric matrix. If the drug is already dissolved in the matrix, it can lead to an initial burst release from the surface. Many factors, such as a change in temperature or pH, the matrix’s composition, and the drug molecule’s size, affect diffusivity [3,4,25]. An eroding polymer matrix is preferred for implantable DDSs. There is no need for DDS retrieval after implantation because the polymeric matrix gets eliminated by erosion. However, the resorbability and toxicity of the degrading products are very important. A drug molecule gets released from DDS only upon the hydrolytic degradation of the matrix. The degradation of DDS depends on the rate of water penetration into the matrix and the kinetics of the hydrolysis process. If water cannot readily penetrate the DDS, surface erosion behavior is observed. Conversely, if water penetrates more rapidly than the degradation rate of the polymeric matrix, bulk erosion is induced. Sometimes degradation products themselves catalyze hydrolysis, leading to autocatalytic degradation [6,25]. Swelling as a drug release mechanism can be used both in polymer matrices and crosslinked polymer networks. The drug is dissolved or dispersed in a matrix with limited diffusivity. Electrostatic and ionic interactions, entropy changes, hydrophilic/hydrophobic interactions, and osmotic stress influence a solvent’s diffusion into the polymer network, leading to solvation and swelling. Generally, the kinetics of drug release from DDSs depends on the surface area and degree of swelling [3,4,25]. Osmosis-mediated controlled DDSs are also used. In osmotic pump-based DDSs, a drug and an osmogen are compacted to form a core compartment, which is enveloped by a semi-permeable membrane. The membrane selectively allows only an inward flow of the solvent under an osmotic gradient. The solvent flow leads to the dissolution of drug molecules, which release out of the system under hydrostatic pressure at a constant rate. However, these systems are highly complicated to fabricate. Moreover, membrane rupture can lead to dose dumping of drugs [4,25].

## 4. Anti-Cancer Drug Delivery Systems Obtained from Biodegradable Polyurethanes

Biodegradable PU-DDSs can be in the form of nano- or microsystems (micelles, nanoparticles, nanocapsules, microspheres, and pellets), membrane systems (films and foams), or matrix systems (gels and scaffolds) (Table 1). The drug release profiles from these DDSs are often discussed in relation to their composition, swelling, initial drug-loading, and degradation rate. The pH and presence of enzymes are also factors that influence drug release kinetics [26].

Biodegradable PU used in PU-DDSs production can be obtained via polyaddition [35,36,37,38,39,40,41,42,43,44,45,46,47,48,49,50,51,52,53,54,55,56,57,58,59,60,61,62,63,64,65,66,67,68] or the polycondensation process [69]. IPDI, HDI, LDI, CDI, HMDI, and BDI were used as IC components in biodegradable PUs synthesis.

A very interesting group of PU-DDSs are the systems responding to physicochemical and biological stimuli.

One strategy focuses on PU-DDSs’ preparation, which characterizes thermal anti-cancer drug release control [35]. For example, thermoresponsive PU-DDSs’ aqueous media were obtained. By increasing the PEG content, an LCST was manifested that could easily be tuned from 30 °C to 70 °C. PU nanoparticles with lower critical solution temperature (LCST) values below the body temperature and temperature-responsive DOX release were characterized by a highly controlled drug release [35]. The amphiphilic aliphatic PU (APU) nanocarriers showed thermoresponsiveness above lower LCST at 41–43 °C, corresponding to cancer tissue’s temperature [69]. The APUs were obtained from L-lysine monomers and 1,12-dodecanediol via a polycondensation process. The obtained nanoparticles accomplished more than 90% of cell death in breast cancer (MCF 7) cells. Moreover, the PU nanoparticles were readily taken up and internalized in the cancer cells [69]. Poly(3-hydroxybutyrate-co-3-hydroxyhexanoate)-based PU thermogels as highly controlled anti-cancer DDSs were also obtained [39]. DTX was released from DDSs with zero-order kinetics for 10 days. DTX-loaded thermogel showed an enhanced anti-melanoma effect on melanoma compared with the free drug and showed no apparent harm to other tissues, including liver, heart, spleen, kidney, and lung tissues [39].

Another strategy is that of pH-stimuli-responsive PU-DDSs [36,47,48,51,52,57]. A very interesting example are PUs obtained from HDI, 2,2-bis(hydroxymethyl) propionic acid, and PEG [42]. An in vitro cellular uptake assay and a Cell Counting Kit-8 assay demonstrated that these DDSs had a higher level of cellular internalization and higher inhibitory effects on the proliferation of human breast cancer (MCF-7) cells than that of pure DOX [42]. Similar DDSs were obtained from IPDI, methoxyl-poly(ethylene glycol) (mPEG), carboxylic acid groups, and piperazine groups. The DDSs released DOX at a controlled rate with a lowering of the pH value [43]. Liu and co-workers obtained pH-stimuli-responsive PU-DDSs (from HDI, PEG, and PCL) characterized by no burst release of DOX [43].

Another way to control the anti-cancer drug release is through redox-sensitive systems [37,45,53,59,65]. Redox-sensitive PU micelles, with tunable surface charge, switch abilities, and crosslinked with pH cleavable Schiff bonds, were obtained. PU micelles with DOX displayed high cytotoxicity against tumor cells [37]. Reduction-responsive micelles based on biodegradable amphiphilic PUs (polyurethane with disulfide bonds and PEG fragments; PEG-PU(SS)-PEG) were obtained by Zhang [38]. Under the influence of the reducing substance glutathione (GSH), the disulfide bond in the main chain broke, triggering the release of the loaded DOX. Cell experiments confirmed that treatment with DOX-loaded PEG-PU(SS)-PEG micelles significantly inhibited the growth of C6 cells compared with that of other groups [38]. GSH-responsive PUs were also described in other papers [44,54]. Similarly, reduction-sensitive PUs were synthesized using a disulfide-containing PCL as the hydrophobic block and a cystamine-functionalized PEG as the hydrophilic block [40]. Under a reductive environment, DOX was released in vitro within 5 h. DOX-loaded PU micelles displayed significant anti-tumor activity [40].

Dual-controlled anti-cancer DDSs were also obtained [41]. One of the more interesting examples of such a system is the pH and redox dual-stimuli-responsive PU micelles prepared from PTMC–SS–PTMC, CDI, and PEOtz–OH (Figure 7). In vitro drug release profiles and cell experiments confirmed that the obtained PU micelles caused controlled DOX release to C6 cells [41]. Another example of a dual system is dendritic PUs synthesized from dipentaerythritol, HDI, mPEG-2000, and glycerol. These obtained PU-DDSs showed excellent pH/ultrasound dual-triggered DOX-release performance [58].

Composite biodegradable DDSs containing PUs were also obtained [49,56,62,68,74]. Obtained waterborne polyurethane (WPU) and chitosan (CS) composite membranes exhibited fine biodegradability, favorable cytocompatibility, and excellent blood compatibility. A cellular uptake assay and CCK 8 assay showed that the DOX was released efficiently from DDSs and taken up by tumor cells [49]. Another important example of systems of this type of DDSs are the nanofibers from cellulose acetate/PU/carbon nanotubes. The obtained results demonstrated the high effects of DOX-loaded nanofibers on the death of LNCaP prostate cancer cells [56]. Farboudi and co-workers obtained a composite DDS (PCL/HDI/PNIPAAm grafted-chitosan core) containing two cytostatics—5-FU and PACL [62]. The drugs were released from nanofibers under an acidic and physiological pH with high control. There was no burst release of PACL and 5-FU from the nanofibers. Incubation of the nanofibers in 4T1 breast cancer cells indicated the good adhesion of cells to the surface of the nanofibers [62].

Other exciting examples are biodegradable PU-DDSs containing superparamagnetic iron oxide nanoparticles (SPION) [50]. It was found that PU micelles in combination with SPION exhibit excellent magnetic resonance imaging (MRI) and the targeting of DOX to the tumor precisely, leading to a significant inhibition of cancer [50].

In summary, many different biodegradable PU-DDSs (or PU-composite DDSs) have been developed, including those which respond to physicochemical and biological stimuli. Despite this, none of the developed DDSs of this type have been clinically applied yet.

## 5. Anti-Cancer Drug Delivery Systems Obtained from Non-Biodegradable Polyurethanes

Non-degradable PUs, used in DDSs technology, are mainly characterized by high biostability, good blood or tissue compatibility, and structural and mechanical strength for long- or medium-term use (e.g., dental and orthopedic implants). The release of drugs from non-degradable PU-DDSs depends mainly on diffusion. The release rate is governed by the thickness and permeability of DDS as well as the drug solubility in the polymer matrix [26].

One of the most interesting examples of non-degradable anti-cancer PU-DDSs are PU foams (synthesized from TDI) containing GEF. Drug-release studies showed a sustained highly controlled release of GEF over nine months (with zero-order kinetics). The developed biomaterial is dedicated for the palliative treatment of bronchotracheal cancer [70]. Another fascinating DDS with very high cytostatic (PACL) release control for 10 days is temperature-responsive PUs (obtained from MDI, PCL, and BDO) [71]. The PU membranes were non-cytotoxic to a broad range of cell lines.

However, in some cases, the use of non-biodegradable or semi-biodegradable PU materials may raise some concerns, mainly possible toxicological problems. Semi-biodegradable implants were obtained using TDI, PEG, and DEG, for example [72]. Some of the developed PU-DDSs were characterized by relatively high CYCLOPHO release control (with near zero-order kinetics). However, the authors did not conduct complete toxicological tests of the obtained biomaterials. Similarly, pH-responsive PU micelles were obtained using MDI [73]. The DOX-loaded PU micelles at pH 6.0 DOX were rapidly released. The released DOX exerted potent anti-proliferative and cytotoxic effects in vitro. However, the authors did not discuss the possible toxicity of the decomposition products of the obtained PUs. Another interesting example of the PU-DDSs is the dual systems (as carriers of PACL and TMZ) constituting the magnetic metal nanoparticles incorporated into poly(acrylic acid) grafted-CS/PU core-shell nanofibers [67]. The obtained results indicated that the synthesized nanofibers could be used for the targeted delivery of anti-cancer drugs with a maximum apoptosis of 49.6% for U-87 MG glioblastoma cells. However, no complete DDSs biodegradation studies or toxicological tests of the carrier degradation products were performed.

The above systems were obtained by a polyaddition process (prepolymer or one-step method).

## 6. Anti-Cancer Polyurethane Prodrug

One of the most interesting types of anti-cancer PU-DDSs are macromolecular prodrugs (macromolecular conjugates). Macromolecular prodrugs are a covalent conjugation of a drug with a polymeric chain. As is well-known, many types of labile chemical bonds are formed between a polymer chain and a drug that are susceptible to enzymatic or hydrolytic degradation (e.g., amide, carbonate, ester, ether, and urethane) [4]. Polyurethane prodrugs (PU-prodrugs) can have varied and complex structures. A drug moiety might be a terminal group of the PU chain, linked to the polymer through a pendant group, or it could also be incorporated into the PU backbone [4,26]. The kinetics of drug release from anti-cancer PU-prodrugs depends on the type of linkage between PU and drug molecules, structure, hydrophilic–hydrophobic properties, and the molecular weights or polydispersity of PU. PUs based on polyester segments degrade faster than PU based on polycarbonate or polyether segments, for example [4,10,26]. The selection of these parameters allows short-, medium-, or long-term anti-cancer PU-prodrugs to be obtained.

One example of a very effective anti-cancer PU-DDSs is 5-FU-PU conjugates obtained from HDI, dihydroxy(polyethylene adipate) (OEDA), and homo- or copolymers of LA and CL [74]. The synthesized PU conjugates are an example of DDS where the drug molecules were incorporated into the polymer chain. Drug-release studies showed the sustained release, and in some cases, highly controlled release, of 5-FU over 35 days (with near zero-order kinetics). It was found that the release of 5-FU from PU-DDSs depended on the nature of oligoester units and consisted of soft and hard segments.

PU conjugates, where molecules of the drug were from the pendant group of the macromolecular chain, were obtained by Qian and co-workers (Figure 8) [57]. The PU-DDSs were obtained from LDI, hydrazine, dihydroxy carboxybetaine, and DOX. The obtained pH-responsive PU-DDSs showed high stability in a physiological environment and continuously released the DOX under acidic conditions. In addition, cytotoxicity studies demonstrated the pure PU carrier to be virtually non-cytotoxic, while the prodrug micelles were more efficient in killing tumor cells.

A novel dendritic polyurethane-based prodrug with a drug content of 18.9% was obtained by conjugating DOX onto the end groups of the functionalized dendritic PU via acid-labile imine bonds [58]. The PU-DDSs showed excellent pH/ultrasound dual-triggered drug release performance, with drug leakage of only 4% at pH 7.4 but a cumulative release of 14% and 88% at pH 5.0 without and with ultrasound, respectively. With ultrasound, the PU micelles possessed greater tumor-growth inhibition than that of free DOX, but without ultrasound, they showed no apparent cytotoxicity on the tumor cells.

The use of PU and anti-cancer drugs conjugates may have certain dangers. During the degradation of the conjugates, drug molecules and various oligomeric fragments containing the active substance molecules are formed. This may pose a toxicological risk due to the lack of complete knowledge about the biological properties of the breakdown products of these conjugates.

## 7. Conclusions, Challenges, and Prospects

According to various global cancer statistics, there were about 20 million new cancer cases and about 10 million deaths last year. Cancer treatment methods include surgery, radiotherapy, chemotherapy, and targeted therapy. Unfortunately, these therapy methods are often limited in many cases of intensely aggressive tumors with a high fatality rate. The attempts to apply polymeric DDSs as a novel and more promising therapy method are increasingly made to improve the cure for and survival rate of cancer patients. DDSs based on PUs have become one of the most interesting directions of research on new anti-cancer drug carriers, further opening a new clinical treatment method for cancer.

What are the main challenges in the technology of new anti-cancer PU-DDSs? Despite intensive research, it has not been possible to resolve the following problems fully: -some of the PUs (mainly based on aromatic isocyanates), products of their degradation, used solvents, etc., may exhibit toxic, irritating, and allergenic properties.-PU nano- and microcarriers have an active and large surface and can “negatively” interact with biomolecules.-The immune system may incorrectly recognize PU-DDSs.-Nano-PU-DDSs have the size of some proteins and can interfere with the transmission of information between cells.-A small number of developed PU-DDSs are characterized by a fully controlled release of the anti-cancer drug.-In many cases, the occurrence of the phenomenon of the drug’s burst release is observed.-Some methods of obtaining PU-DDSs are multi-stage and complex.-The cost of raw materials and technologies for obtaining PU-DDSs is, in many cases, high.

Although many developed anti-cancer PU-DDSs have been used in pre-clinical trials, no obtained system has been approved for commercial use. It is worth considering that most of these studies are in early stages, and their clinical effects must be further verified. However, positive and prospective biological and pharmaceutical test results in many cases indicate the need for further work on these types of DDS, and they offer hope for their quick practical application in cancer therapy.

## Figures and Tables

**Figure 1 ijms-23-08181-f001:**
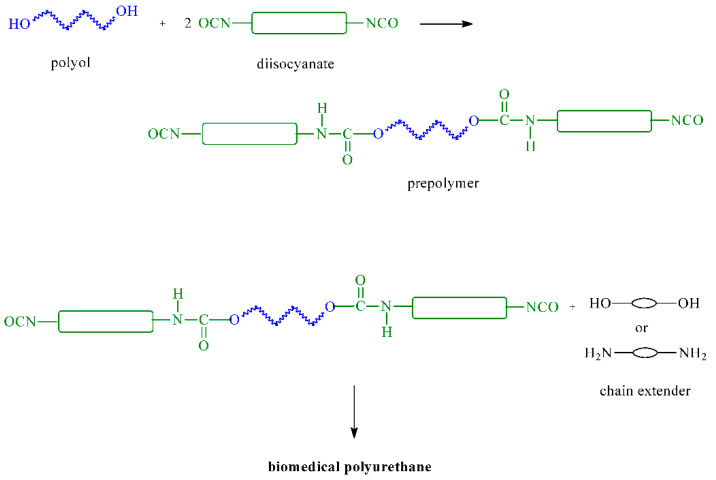
Synthesis of biomedical polyurethanes with the prepolymer method.

**Figure 2 ijms-23-08181-f002:**
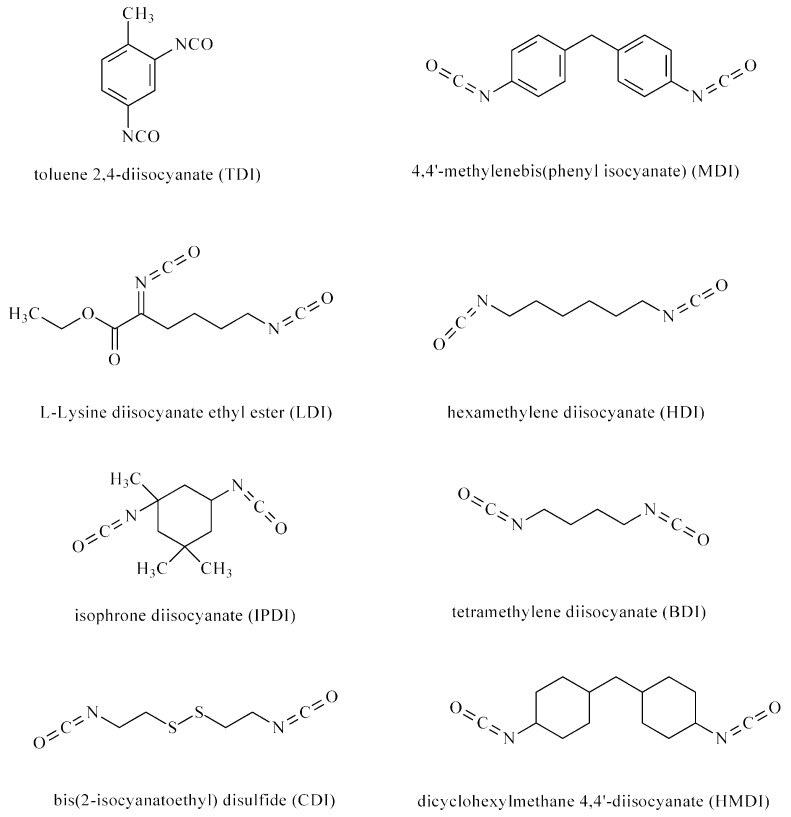
Diisocyanates used for the synthesis of biomedical polyurethanes.

**Figure 3 ijms-23-08181-f003:**
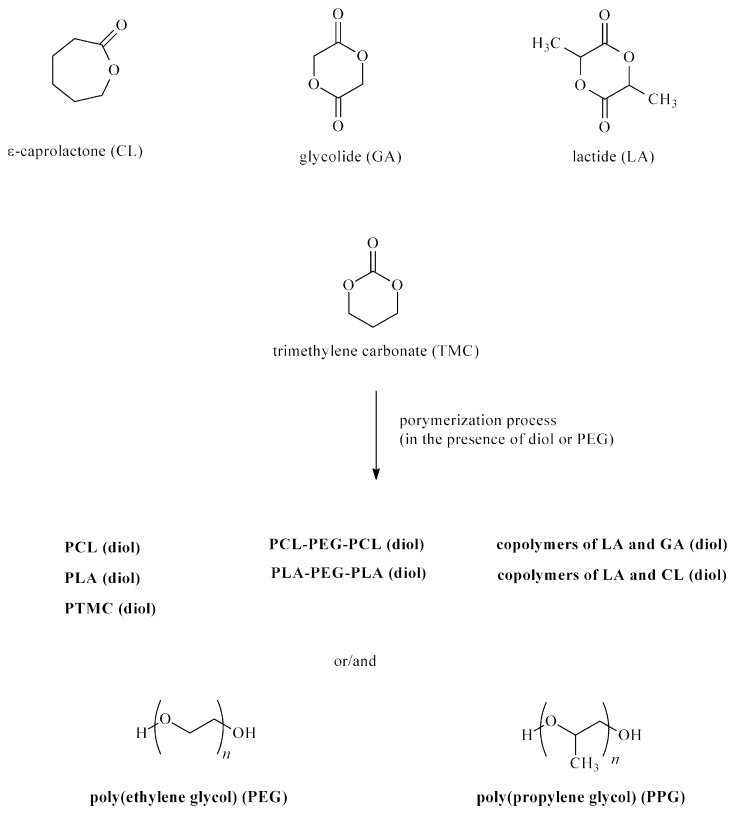
Polyols used for the synthesis of biomedical polyurethanes.

**Figure 4 ijms-23-08181-f004:**
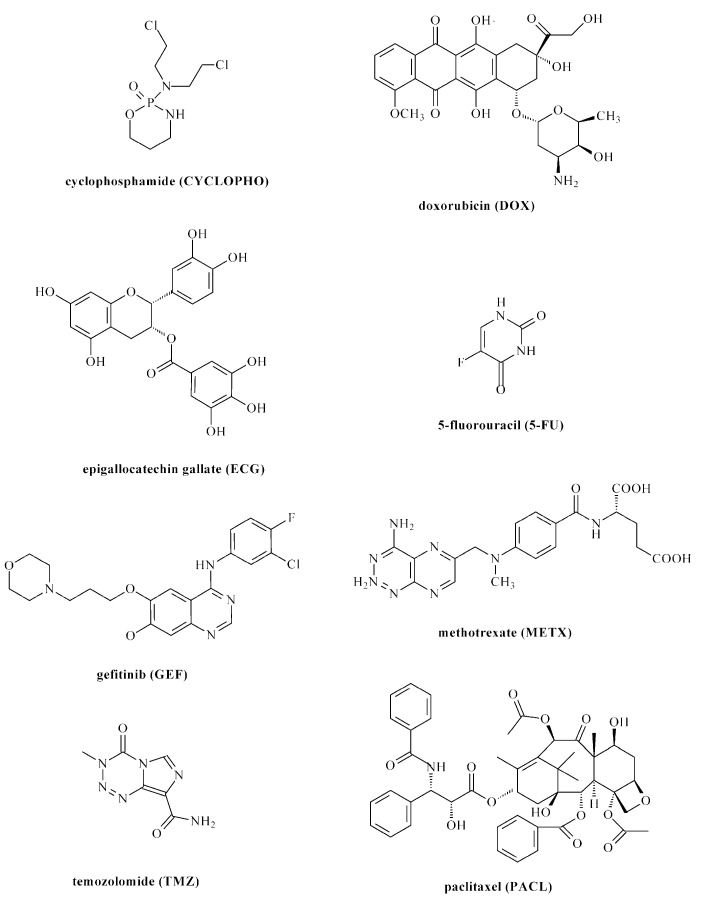
Anti-cancer drugs used in the technology for polyurethane drug delivery systems.

**Figure 5 ijms-23-08181-f005:**
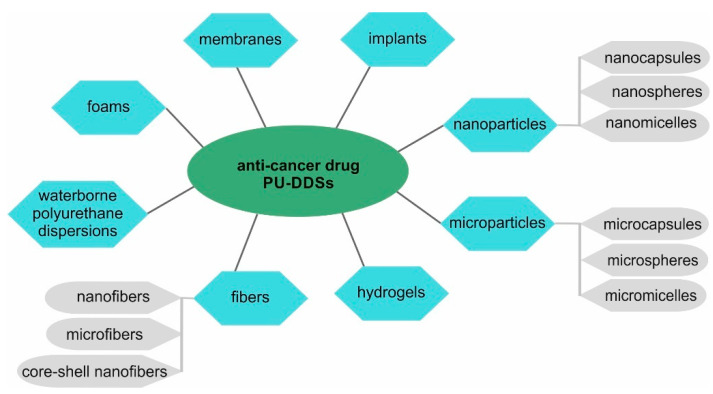
Types of polyurethane drug delivery systems for anti-cancer drugs.

**Figure 6 ijms-23-08181-f006:**
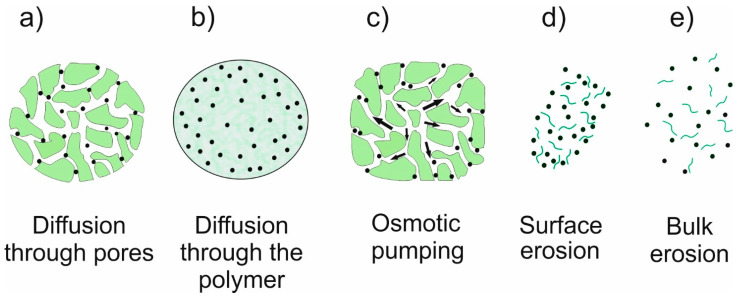
The drug release mechanisms involved in different polymeric drug delivery systems.

**Figure 7 ijms-23-08181-f007:**
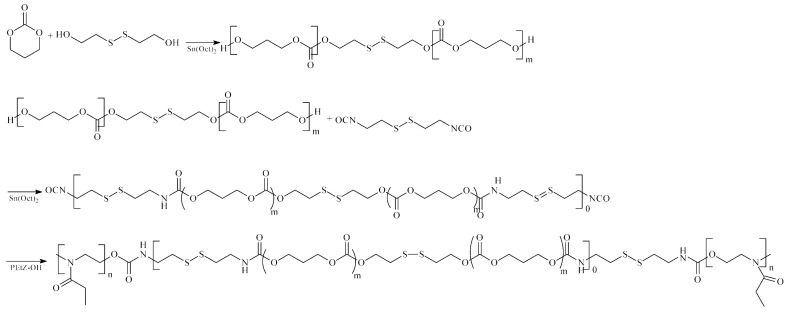
Synthesis of the pH and redox dual stimuli-responsive polyurethanes.

**Figure 8 ijms-23-08181-f008:**
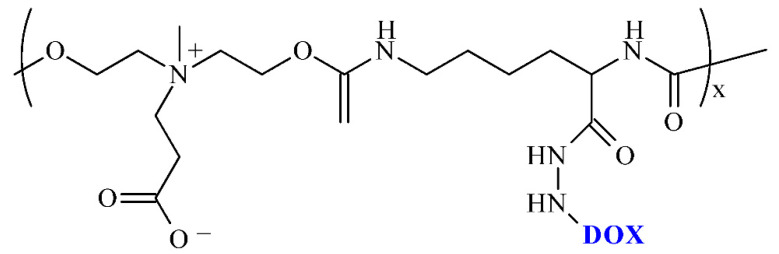
Structure conjugates of polyurethane and doxorubicin.

**Table 1 ijms-23-08181-t001:** Polyurethane drug delivery systems.

Drug/Drugs	Type of PUs or Composites	Type of DDSs	Main Conclusions	Ref.
DOX	PEG-1500/bis-MPA/IPDI	nano- and microparticles/injectable carriers	- Thermoresponsive PUs manifested an LCST that could be easily tuned from 30 °C to 70 °C by increasing the PEG content. - Temperature-responsive PU nanoparticles were characterized by a highly controlled DOX release.	[35]
DOX	HDI/PCL/PEG	microcapsules	- pH-sensitive PU-DDSs were easily internalized by BGC 823 and Hela cells. - PU-DDSs were characterized by a highly controlled drug release.	[36]
DOX	PU-SS-COOH: PEG-1000/PCL-2000/HDI/CYS/DMPA; PU-SS-COOH-NH_2_: PEG-1000/PCL-2000/HDI/CYS/DMPA/1,6-diaminohexane	micelles	- The DOX release rate from the redox-sensitive PU micelles was controlled by the addition of GSH. - DOX-loaded PU micelles displayed high cytotoxicity against tumor cells.	[37]
DOX	LDI/PEG-PU(SS)-PEG/	micelles	- DOX-loaded PU micelles had good stability under the extracellular physiological environment, but the drug was released quickly under the intracellular reducing conditions. - DOX-loaded PU micelles had a high in vitro anti-tumor activity in C6 cells;	[38]
DOX	PHBHx/PEG-2000/PPG-2050/HDI	thermogel	- DOX was released from thermogel with zero-order kinetics during 10 days. - DOX-loaded thermogels showed an enhanced anti-melanoma effect on melanoma solid tumors and no apparent harm to other tissues, including liver, heart, spleen, kidney, and lung tissues.	[39]
DOX	LDI/mPEG-OH-5000/PCL; PCL obtained form ε-CL to 2,20-dithiodiethanol	micelles	- DDSs for osteosarcoma therapy were obtained. - In vitro, DOX-loaded PU micelles displayed significant anti-tumor activity, which was comparable with that of free DOX, against Saos–2 cells.	[40]
DOX	PTMC-SS-PTMC/CDI/PEOtz-OH	micelles	- The pH and redox dual stimuli-responsive PU micelles were characterized by controlled DOX release to C6 cells.	[41]
DOX	HDI/2,2-bis(hydroxymethyl) propionic acid/PEG; amphiphilic PUs with carboxyl pendent groups	nanoparticles	- pH-sensitive PU nanoparticles (NP) had a higher level of cellular internalization and higher inhibitory effects on the proliferation of human breast cancer (MCF-7) cells than that of pure DOX.	[42]
DOX	IPDI/methoxyl-poly(ethylene glycol) (mPEG)/carboxylic acid/piperazine	micelles	- The drug release of DOX-loaded PS–PU micelles showed an obvious step-up with the reducion of the pH. - The charge-reversal property improved the cellular uptake behavior and intracellular drug release in both HeLa cells and MCF-7 cells.	[43]
DOX	mPEG-5000/HDI/trimethylolpropane/bis(2-hydroxyethyl) disulfide	core-shell nanogels	- GSH-responsive PU-based core-shell nanogels with hydrophilic mPEG shell were prepared. - GSH triggered the nanogel swelling and accelerated the loaded drug release in PBS (pH = 7.4).	[44]
DOX	poly(2-oxazoline)s/PLA-SS-PLA/LDI	micelles	- The release of the drug was stimulated in an acidic and reductive environment. - The DOX-loaded PU micelles had high activity against C6 (rat glioma cells) cells.	[45]
DOX	PEG-2000/HDI and PCL-2000/PEG-2000/HDI	nanomicelles	- PU micelles had higher cytotoxicity compared with pure DOX. - The obtained micelles had better tumor inhibition ability and safety than that of pure DOX. - DOX micelles had almost no burst release of the drug in a pH 7.4 environment.	[46]
DOX	mPEG-1000 (or PEG-2000)/poly(1,3-propylene succinate) diols (PPS)/IPDI	micelles	- The enzymatic degradation of the micelles for 8 weeks under the physiological environment revealed that the degradation mainly occurred at the ester group of PPS blocks. - A cytotoxicity test proved that the PU micelles were non-toxic, while the DOX-loaded micelles showed concentration-dependent cytotoxicity to HeLa cells.	[47]
DOX	PLA-SS-PLA/LDI/PEG	micelles	- DOX was released quickly under intracellular reducing conditions. - CCK-8 assays showed that DOX-loaded PU micelles had high in vitro anti-tumor activity in C6 cells.	[48]
DOX	WPU/CS	membranes	- Waterborne polyurethane (WPU) and chitosan (CS) composite membranes exhibited fine biodegradability, favorable cytocompatibility, excellent blood compatibility, and a well-sustained release effect manifested in slow release, stability, and no sudden releases. - DOX can be released efficiently from the drug-loading matrix and taken up by tumor cells.	[49]
DOX	mPEG-1900/PCL/LDI; PUs with benzoic-imine linkage	micelles	- The cleavage of PEG corona bearing a pH-sensitive benzoic-imine linkage could act as an on–off switch, which is capable of activating clicked targeting ligands under an extracellular acidic condition, followed by triggering a core degradation and payload release within tumor cells.	[50]
DOX	polycondensation products of ortho ester-based diols and HDI (or HMDI)	microparticles	- pH-sensitive POEUs NP were stable at physiological condition (7.4), were characterized by an accelerated degradation at a mildly acidic pH (5.0), an effective intracellular delivery of DOX, and high anti-tumor activity against 2D monolayer cells in vitro, and significantly enhanced the penetration of DOX into 3D multi-cellular tumor spheroids.	[51]
DOX	polycondensation product of terephthalilidene-bis(trimethylolethane) and LDI (and next termination process with allyl alcohol)	nanomicelles	- In vitro DOX was released from obtained nanomicelles in a controlled and pH-dependent manner. - DOX-loaded PU micelles had high in vitro anti-tumor activity in both RAW 264.7 and drug-resistant MCF-7/ADR cells.	[52]
DOX	*trans*-4,5-dihydroxy-1,2-dithiane (O-DTT)/HDI/mPEG	nanomicelles	- DOX-loaded PU micelles exhibited high anti-tumor efficacy in vivo with reduced toxicity.	[53]
DOX	PEG-2000/bis-1,4-(hydroxyethyl) piperazine (HEP)/O-DTT/HDI	nanomicelles	- PU micelles tended to decompose under a weakly acidic environment or in the presence of an intracellular reducing agent (GSH).	[54]
DOX	LDI/PDO/PEG/PCL/folic acid (FA)	nano- and micelles	- FA-conjugated PU micelles displayed a sustained DOX release, preferential internalization by human epidermoid carcinoma cell line (KB cells), and pronounced cytotoxicity compared with PU micelles without FA.	[55]
DOX	PCL/poly (tetramethylene ether) glycol/HDI	cellulose acetate/PU/carbon nanotubes/composite nanofibers	- The synergic effects of composites and DOX-loaded nanofibers on the death of LNCaP prostate cancer cells were observed.	[56]
DOX	LDI/hydrazine/dihydroxy carboxybetaine	conjugates/nano- and micromicelles	- pH-responsive PU-DDSs showed high stability in a physiological environment and continuously released DOX under acidic conditions. Carrier was virtually non-cytotoxic, while the prodrug micelles were more efficient in killing tumor cells.	[57]
DOX	Dipentaerythritol/HDI/mPEG-2000/glycerol	conjugates/nanomicelles/dendritic PU	- PU-DDSs showed excellent pH/ultrasound dual-triggered drug release and tumor growth inhibition performance.	[58]
DOX and PACL	PLA-SS-PLA/IPDI/PEG	micelles	- PACL release from DDSs was significantly accelerated by redox stimuli. - PU micelles showed high cytotoxicity against HepG2 tumor cells.	[59]
ECG	MEG/BDO/PEG-200/HDI/IPDI	microparticles	- The in vitro cytotoxic effect of obtained PU loaded with ECG on human pharyngeal carcinoma cells (Detroit 562) and squamous cell carcinoma (SCC-4) was observed.	[60]
5-FU	HDI/PEG-650 or -1250 or -1500 or -2000/1,2−DAE or 1,6-DAH or 1,4-DAB or 1,8-DAO/L-LYS	WPU	- WPU were characterized with highly controlled drug-released kinetics. - The 5-FU release rate was easily controlled in relation to the chain length of the chain extender and Mw of PEG.	[61]
5-FU and PACL	(PCL/HDI)/PNIPAAm grafted-chitosan core-shell nanofibers	core-shell nanofibers	- PACL and 5-FU were released from nanofibers under a acidic and physiological pH with high control (and no burst release of drugs). - The minimum increase in tumor volume was obtained using PACL and 5-FU loaded-nanofibers coated by magnetic gold nanoparticles.	[62]
METX	PCL-b-PEG-b-PCL/BDI/ L-glutathione oxidized	films	- In some cases, the drug was released with sustained highly controlled kinetics over a period of 96–144 h (with near zero-order kinetics).	[63]
PACL	L-LYS-GQA/L-LYS-ABA-ABA tripeptide/HPCL/HPEG/LDI/PDO	nanomicelles	- Nanocarriers improved cellular internalization and triggered intracellular PACL release in response to acidity within tumor cells.	[64]
PACL	PEG-1000/PCL-2000/LDI/BDO/CYS or PEG-1000/PCL-2000/LDI/MDEA/BDO or PEG-1000/PCL-2000/LDI/CYS/MDEA	micelles	- PACL was released from PU micelles within 48 h in response to acidic and reductive stimuli;. - Intracellular release of anti-cancer drug and internalization into H460 cancer cells was evidenced.	[65]
PACL	PCL-co-PEG/HMDI	nanoparticles	- A biodistribution study of healthy mice evidenced no relevant differences between the commercial drug (Taxol) and obtained NP forms of PACL.	[66]
PACL and TMZ	PU purchased from Lubrizol Co	magnetic particles incorporated into nanofibers	- Magnetic MIL-53 nanometal organic framework particles incorporated into poly(acrylic acid) grafted-CS/PU core-shell nanofibers were obtained. - Nanofibers induced maximal apoptosis of U-87 MG glioblastoma cells.	[67]
TMZ	PCL/HDI/BDO	- NP incorporated into nanofibers; - gold-coated NP-loaded PU nanofibers;	- NP (CS/TMZ) incorporated into nanofibers (PU/TMZ) and gold-coated (CS/TMZ) NP-loaded PU nanofibers were obtained. - The obtained nanofibers inhibited the growth of U-87 MG human glioblastoma cells. - Sustained TMZ release from DDSs for 30 days with the zero-order kinetic model was achieved.	[68]
DOX	polycondensation products of multi-functional L-lysine monomers/1,12-dodecanediol	nanomicelles	- The amphiphilic aliphatic PU (APU) nanocarriers showed thermoresponsiveness above the lower LCST at 41–43 °C corresponding to cancer tissue temperature. - The DOX-loaded APU nanoparticles accomplished more than 90% cell death in breast cancer (MCF 7) cells.	[69]
GEF	TDI/unknown polyol/unknown cross-linker (Vysera Biomedical Ltd.); GEF-loaded PLGA-based microspheres	PU foams either as micronized drug or as GEF-PLGA microspheres	- The coating of drug-eluting stents for the palliative treatment of bronchotracheal cancer was obtained. - The drug was released with sustained highly controlled kinetics of GEF over a period of nine months (with zero-order kinetics).	[70]
PACL	MDI/PCL-4000/BDO	membrane	- Temperature-responsive PU membranes exhibited a switching temperature at 44 °C. - Below the switching temperature, shrunken free volume within the polymeric matrix prevented the incorporated PACL from diffusing out; upon heating above the switching temperature, the PU membranes rapidly switched on, allowing dramatically accelerated drug diffusion.	[71]
CYCLOPHO	TDI/PEG-600 (or -1500 or -3500)/DEG	implant	- High control of the CYCLOPHO release from PU-DDSs - Reduced toxic action of PU-DDSs compared with drug injections (in vivo tests—rats)	[72]
DOX	MDI/PPG-N_3_/PPEG-2000 or PPEG-4000	micelles	- At pH 6.0, DOX was rapidly released from pH-responsive PU micelles. - Released DOX exerted potent anti-proliferative and cytotoxic effects in vitro. - Micelles safely and efficiently delivered DOX into the cell nuclei.	[73]
5-FU	PCL (or PLA, CL/LA copolymers)/HDI	conjugates	- In some cases, a highly controlled release of 5-FU over a period of 35 days was observed (with near zero-order kinetics).	[74]

## Data Availability

Not applicable.

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
