# Peer review of "Biomedical Polyurethanes for Anti-Cancer Drug Delivery Systems: A Brief, Comprehensive Review"

_ijms, 2022, doi:10.3390/ijms23158181_

Round 1

Reviewer 1 Report

This review summarizes well the work that has been done on PU drug delivery systems for applications in cancer therapy. The authors describe well the different categories of PU systems and the various drugs that they carry. They also describe well the PU syntheses, with appropriate synthetic schemes.

Concerning the tables, I think that Table 1 is not necessary: these drugs are already known, and it should be enough to just say that they are all anti-cancer drugs with different modes of actions and give references 26-33 in the text.

Table 2 is nice but really too long. The main conclusions are too long for each reference. It should not be one row per reference. The references should be grouped in subcategories so that they are several references per row in the table.

In Figure 6, the caption should say "in different polymeric drug delivery systems."

The conclusion is appropriate.

Author Response

No comments.

Reviewer 2 Report

The manuscript titled “Biomedical polyurethanes for anticancer-drug delivery system: a brief” by Sobzak, M.; et al. is a necessary review work where the authors state the most recent and promising strategies based on polyurethanes chemistry to fight against cancer diseases. Multiple isocyanate crosslinker molecules are presented being their respective application on oncology therapies addressed. This scientific work is clear which will significantly aid to other researchers in the same field and other stakeholders to design more efficient and effective carcinogenic polyurethane (Pus)-based treatments specially focused on drug delivery systems (DDSs). The presence of tables also helps to better visualize the prospect of this research topic to potential readers. The present review not only can be implemented in the cancer healthcare but also to other research topics like dental implants, food packaging, binders, among others. The shown findings may be relevant for the examined field. The results achieved are well-discussed during the main body of the reported manuscript. The scientific paper is well written. In my opinion the present manuscript is innovative and the methodological approached used matches with the scope of International Journal of Molecular Sciences. For the above described reasons, I recommend the publication in International Journal of Molecular Sciences once the following remarks will be fixed:

--------

TITLE

I may suggest add a small statement at the end of the title like “comprehensive review/outlook/overview”. Thus, the final title version will be “Biomedical polyurethanes for anticancer-drug delivery system: a brief comprehensive review” (or outlook/overview).

--------

KEYWORDS

The keywords selected by the authors are right. Nevertheless (and taking into account that the maximum number of allowed keywords is ten) I may introduce one additional keyword like “polyurethane chemistry” or “polyurethane reactivity”.

--------

1. INTRODUCTION

The introduction section is accurate. Even if in next sections authors state the properties of biomedical polyurethanes it would be desirable also to include their general properties in Introduction section (like excellent elongation, elasticity and tensile strength performance, corrosion resistant, high gloss and weathering resistance) [1,2,3].

[1] Brzeska, J.; et al. Morphology and Physicochemical Properties of Branched Polyurethane/Biopolymer Blends. Polymers 2019, 12, 16. https://doi.org/10.3390/polym12010016.

[2] Jia, Z.; et al. Preparation and Mechanical-Fatigue Properties of Elastic Polyurethane Concrete Composites. Materials 2021, 14, 3829. https://doi.org/10.3390/ma14143839.

[3] Borowicz, M.; et al. Effect of New Eco-Polyols Based on PLA Waste on the Basic Properties of Rigid Polyurethane and Polyurethane/Polyisocyanurate Foams. Int. J. Mol. Sci. 2021, 22, 8981. https://doi.org/10.3390/ijms22168981.

--------

2. SYNTHESIS AND PROPERTIES OF BIOMEDICAL POLYURETHANES USED IN ANTI-CANCER DRUG DELIVERY SYSTEMS.

 “The different types of ICs (aromatic, aliphatic, cycloaliphatic) are used in BioPUs synthesis: (…)”. Here, it would be opportune to discuss about the potential use of isocyanate crosslinker molecules containing cleaved aromatic phenolic groups because it has been reported their proven ability not to present adverse effects on cell proliferation [4].

[4] Phung Hai, T.A.; et al. Renewable Polyurethanes from Sustainable Biological Precursors. Biomacromolecules 2021, 22, 1770-1794. https://doi.org/acs.biomac.0c01610.

--------

3. POLYURETHANE ANTI-CANCER DRUG DELIVERY SYSTEMS

(Optional) Chemical compounds from Figure 3, being also extensible to Figures 7 and 8, are taken by screenshots from the commercial supplier website. This fact makes these images partially blurred. I may suggest to the authors the use of softwares like ChemDraw to prepare these chemical compound images.

Then, Figure 5 should be slightly enlarged to better visualize the content.

--------

6. CONCLUSION, CHALLENGES, AND PROSPECTS

First, I may change the title of this sub-section by “conclusions, challenges, and prospects”.

The author perfectly explains the future perspectives of polyurethanes in cancer drug delivery systems field. Nevertheless, it would be advisable to introduce some additional isocyanate crosslinker molecules used for other applications and which eventually could be employed in DDSs field. In this framework, linker molecules like 3-(triethoxysilyl)-propyl-isocyanate (TEPIC) conjugated with nanoparticles [5] or cellulose nanocrystals functionalized with p-maleidophenyl isocyanate (PMPI) [6] have interesting applications in the development of antimicrobial properties [7] and production of more durable food packaging [8]. Moreover, PUs can be also used to design high-throughput conductive composites [9].

[5] Holló, B.B.; et al. Synthesis, spectroscopic and termal characterization of new metal-containing isocyanate-based polymers. J. Therm. Anal. Calorim. 2017, 132, 1. https://doi.org/10.1007/s10973-017-6904-1.

[6] Marcuello, C.; et al. Langmuir-Blodgett Procedure to Precisely Control the Coverage of Functionalized AFM Cantilevers for SMFS Measurements: Application with Cellulose Nanocrystals. Langmuir 2018, 34, 9376-9386. https://doi.org/10.1021/acs.langmuir.8b01892.

[7] Liu, Y.; Preparation and Enhanced Antimicrobial Activity of Thymol Immobilized on Different Silica Nanoparticles with Application in Apple Juice. Coatings 2022, 12, 671. https://doi.org/10.3390/coatings12050671.

[8] Berzin, F.; et al. Influence of the polarity of the matrix on the breakage mechanisms of Lignocellulosic fibers during twin-screw extrusion. Polym. Compos. 2020, 41, 1106-1117. https://doi.org/10.1002/pc.25442.

[9] Choi, S.; et al. Revised Manuscript with Corrections: Polyurethane-Based Conductive Composites: From Synthesis to Applications. Int. J. Mol. Sci. 2022, 23, 1938. https://doi.org/10.3390/ijms23041938.

Finally, recently it has been reported novel DDSs based on collagen and cellulose crosslinked by Pus [10].

[10] Anghel, N.; et al. Transcutaneous Drug Delivery Systems Based on Collagen/Polyurethane Composites Reinforced with Cellulose. Polymers 2021, 13, 1845. https://doi.org/10.3390/polym13111845.

--------

REFERENCES

All bibliography citations are in the proper format of International Journal of Molecular Sciences.

--------

OVERVIEW AND FINAL COMMENTS

The submitted work is well-designed and the gathered results are interesting for the biomedical and clinical fields in special related for oncology drug delivery applications. For this reason, I will recommend the present scientific manuscript for further publication in International Journal of Molecular Sciences once all the aforementioned suggestions will be properly fixed.

Author Response

No comments.
